# Astragalus polysaccharides and astragaloside IV alleviate inflammation in bovine mammary epithelial cells by regulating Wnt/β-catenin signaling pathway

**Jiaqi Fan, Fang Jia, Yang Liu, Xuezhang Zhou**◯*

Key Laboratory of the Ministry of Education for the Conservation and Utilization of Special Biological Resources of Western China, Ningxia University, Yinchuan, Ningxia, China

* zhouxuezhang@nxu.edu.cn

**Data Availability Statement:** All data generated or analyzed during this study are included in this published article.

## Abstract

The Wnt/β-catenin signaling regulates cell renewal and repair and is closely associated with inflammation. Astragalus polysaccharides (APS) and astragaloside IV (AS-IV), which are the main active substances extracted from *Radix Astragali*, protect cells by regulating Wnt signaling in cells, exerting antiinflammatory, antioxidant, and antistress effects. However, the mechanisms by which APS and AS-IV interact with Wnt signaling to achieve their therapeutic effects in bovine mammary epithelial cells (BMECs) are not understood. In this study, we used lipopolysaccharide (LPS)-stimulated BMECs as an in vitro model of inflammation to investigate the effects of APS and AS-IV on Wnt signaling in inflamed BMECs. Drug concentrations were screened using the CCK-8 method, the effect on protein expression was analyzed using immunoblotting, the effect on inflammatory factors using enzyme-linked immunosorbent assay, and the effect on oxidative factors using enzyme labeling and flow cytometry. LPS activated the expression of inflammatory and oxidative factors in cells and inhibited Wnt/β-catenin signaling. APS and AS-IV antagonized the inhibitory effect of LPS, protecting BMECs. They inhibited the expression of the IL-6, IL-8, and TNF-α inflammatory factors, and that of the MDA oxidative factor, and activated Wnt signaling in LPS-stimulated BMECs. Silencing of β-catenin abolished the protective effect of APS and AS-IV against LPS-stimulated BMECs. Thus, APS and AS-IV mediate protective effects in inflammatory BMECs model through activation of the Wnt signaling pathway. Wnt signaling pathway is one of the targets of the inhibitory effects of APS and AS-IV on inflammation.

## Introduction

Bovine mammary epithelial cells (BMECs) are lactating and immune-related cells of the mammary tissue that have been shown to respond immunologically to bacterial invasion and toxin attacks [1]. In particular, BMECs are the first targets of attack by bacteria and fungi invading the mammary tissue through the ducts. There are more than 150 different types of pathogens in the environment and milking equipment that can cause mastitis [2]. Mastitis, a recurring

**Funding:** This work was supported by a grant from the Natural Science Foundation of Ningxia Hui Autonomous Region (2020AAC03111). The funders had no role in study design, data collection and analysis, decision to publish, or preparation of the manuscript.

**Competing interests:** The authors declare that they have no conflict of interest.

and complex inflammatory disease of dairy cattle, causes 35 billion dollars in economic losses to worldwide husbandry each year, by significantly reducing milk production and increasing farming costs and culling rates [3–5].

Bacterial mastitis infections by gram-negative bacteria lead to the production of a large number of toxins and lipopolysaccharides (LPS), causing a strong immune response [6]. Several studies have used LPS to stimulate cells or the mammary tissue in mice to promote the release of inflammatory factors, induce oxidative stress, and establish models of inflammation [7]. LPS has been shown to regulate signaling pathways in cells by binding to cell surface receptors, triggering the expression of inflammation-related factors [3, 8]. Accordingly, both the NF-κB and MAPK signaling pathways are known to be positively associated with cellular inflammation, whereas an increasing number of studies has identified disruptions in the Wnt/β-catenin signaling pathway that accompany inflammatory responses in a variety of diseases [9–11]. However, whether the Wnt signaling pathway is positively or negatively associated with inflammation remains controversial [9, 10]. The Wnt/β-catenin signaling pathway, which has a regulatory role in cell growth, reproduction, migration, differentiation, invasion, apoptosis, and other life processes, is ubiquitous in various animal cells [12, 13]. In a variety of diseases, such as cancer, fibrosis, autoimmune diseases, and metabolism-related disorders, not only Wnt signaling is aberrantly expressed, but is also accompanied by a sustained inflammatory response [14, 15]. A variety of natural products have been used to alleviate mastitis, but the lack of pathogenetic studies has limited their use [5, 11].

Astragalus polysaccharides (APS) and astragaloside IV (AS-IV) are the main active components of *Radix Astragali* (RA) with antioxidant, antiinflammatory, antitumor, antiviral, anti-stress, and cytoprotective effects [16, 17]. Moreover, they have been reported to regulate a variety of signaling pathways in cells, including the Wnt signaling pathway [18, 19]. In a murine rheumatoid arthritis model, AS-IV was shown to inhibit the expression of nitric oxide (NO), tumor necrosis factor-α (TNFα), and interleukin-1β (IL-1β) [20]. In a model of LPS-induced inflammation in cardiomyocytes, APS inhibited the expression of inflammatory factors by affecting the NF-κB and PI3K/AKT signaling pathways [21]. Of note, AS-IV was found to reduce the expression of β-catenin in mouse keratinocytes, promote cell proliferation, and facilitate cell migration to inhibit ulcerogenesis [16]. Moreover, APS inhibits breast cancer cell migration and invasion by regulating epithelial-mesenchymal transition via the Wnt/β-catenin signaling pathway [18]. These results suggested that APS and AS-IV regulate the expression of Wnt signaling and inflammatory factors in cells. However, the effect of Wnt signaling on mastitis has not yet been investigated. The aim of this study was to investigate the potential effect of APS and AS-IV interventions in the inflammatory inhibition and regulation of Wnt/β-catenin signaling in LPS-induced BMECs, and elucidate the underlying mechanisms. Here, we demonstrated that APS and AS-IV exerted cytoprotective effects on BMECs through the negative regulation of inflammation and Wnt/β-catenin signaling.

## Materials and methods

### Cell culture and treatment

BMECs were preserved in our laboratory, the cell line (BMECs) was obtained from Shandong Agricultural University, Tai'an, China [22]. BMECs were incubated in Dulbecco's modified Eagle's medium (DMEM) (Biological Industries, Israel) supplemented with 10% FBS (v/v) (Gibco, Waltham, MA, USA), 5 μg/mL insulin (Sigma-Aldrich, St. Louis, MO, USA), 5 μg/mL transferrin (Sigma-Aldrich), 1 μg/mL hydrocortisone (Sigma-Aldrich), 1 μg/mL corporin (Sigma-Aldrich), and 100 U/mL penicillin-streptomycin (Sigma-Aldrich) in an environment of 5% $CO_2$ and 95% air at 37°C. Cells were cultured for 24 h before experimental treatment.

When growing cell monolayers reached 80–90% confluence, BMECs were digested with 0.025% Trypsin-EDTA (Solarbio, Beijing, China). AS-IV ($C_{41}H_{68}O_{14}$; molecular weight, 784; HPLC > 98%) (Solarbio) was dissolved in absolute ethanol and diluted in DMEM; the final concentration of absolute ethanol in each group did not exceed 1% (v/v). APS (UV ≥ 90%) (Solarbio) and LPS (Sigma-Aldrich) were also diluted in DMEM.

## Cell viability assay

To assay cell viability, BMEC cells were inoculated in 96-well plates at $1 \times 10^4$ cells/well, with each experimental group being set up in 6 replicates. Cells were incubated for 24 h and grown to 90% confluence for subsequent experiments. Cells were washed twice with 100 μL PBS (Solarbio). For the LPS experiment, cells were divided into 6 groups, that is, 0, 0.1, 0.5, 1, 2.5, and 5 μg/mL LPS-stimulated groups, and incubated for 24 h. For the AS-IV experiment cells were divided into 9 groups, 0, 12.5, 25, 50, 75, 100, 150, 200, and 300 μg/mL AS-IV-stimulated groups, and incubated for 1 h. For the APS experiment cells were divided into 6 groups, 0, 0.1, 0.25, 0.5, 1, and 2 mg/mL APS-stimulated groups, and incubated for 1 h. Cells were then washed twice in PBS, and 100 μL DMEM containing 10% cell counting kit (CCK)-8 solution (Dojindo Laboratories, Kumamoto, Japan) was added to each well. The 96-well plates were incubated for 3 h at 37˚C in the dark, after which the absorbance was measured at 450 nm using a spectrophotometer (Bio-rad, USA).

## Western blot analysis

Total protein was isolated from BMECs using a tissue protein extraction kit (Nanjing KeyGEN Biotech. Co. Ltd., Nanjing, China), and protein concentration was measured using a BCA assay kit (Nanjing KeyGEN). The protein loading volume was adjusted to 30 μg/well. Proteins were subjected to gel electrophoresis (SDS-PAGE, 12.5%) and transferred to polyvinylidene fluoride (PVDF) membranes (Sigma-Aldrich). PVDF membranes were incubated with β-catenin, GSK3β, phospho-GSK3β (Bioss, Beijing, China), axin-1, TCF-4, cyclinD1 (Biosciences, Shanghai, China), and β-actin (Abcam, Cambridge, UK) antibodies for 12 h at 4˚C and washed 3 times with TBST, before incubation with horseradish peroxidase-labeled secondary antibodies (Abcam) for 2 h at 25˚C and a final thrice wash with TBST. Exposure and capturing of images were performed using an ECL chemiluminescent color development solution (Shanghai Epizyme Biomedical Technology Co., Ltd, Shanghai). Grayscale values were analyzed using the ImageJ software.

## Screening siRNA

The cell suspension of BMECs was collected and seeded in a 6-well plate at $2 \times 10^5$ cells/well and cultured for 12 h. Then, siRNA was added when cell density reached 30–40%. The siRNA was designed and synthesized by Shanghai Jima Pharmaceutical Company; its sequence is shown in Table 1. The siRNA was dissolved in DEPC water to prepare a 20 μM solution. Subsequently, 10 μL siRNA was mixed with 10 μL Zeta Life Advanced Transfection Reagent (Zeta Life, USA), and placed at 25˚C 15 min. The cell supernatant was discarded, 10 μL of transfection reagent was added to each well, followed by the addition of basic culture medium to a volume of 1 mL, and the mixture was slightly shaken to mix. Treatments were divided into 4 groups. C, control group; NC, negative control-siRNA group; 1225, 1225-siRNA group; and 1990, 1990-siRNA group. Western blot analysis was performed to detect the protein expression of β-catenin and β-actin.

**Table 1. siRNA sequence.**

| siRNA | | Sense (5'-3') | Antisense (5'-3') |
|---|---|---|---|
| siRNA | CTNNB1-1225 | GGGCUCAGAUGAUAUCAAUTT | AUUGAUAUCAUCUGAGCCCTT |
| siRNA | CTNNB1-1990 | GGACAAGCCACAGGAUUAUTT | AUAAUCCUGUGGCUUGUCCTT |
| NCsiRNA | Negative control | UUCUCCGAACGUGUCACGUTT | ACGUGACACGUUCGGAGAATT |

## Real-time qPCR

Total RNA was extracted using the TRIzol reagent (Invitrogen, Carlsbad, CA, USA). Then, cDNA was synthesized by reverse transcription according to the procedure described in the HiScript II Q Select RT SuperMix for qPCR kit (Vazyme Biotech Co. Ltd, Nanjing, China). Fluorescent quantitative PCR was performed according to the procedure described in the ChamQ Universal SYBR qPCR Master Mix kit (Vazyme). The $2^{-\Delta\Delta Cq}$ method was used to analyze the expression of target genes, and β-actin was used as an internal reference [22]. Primer design for qPCR is shown in Table 2 [22].

## ELISA assay

The supernatant of samples from each experimental group was collected and centrifuged for 10 min at 3000 $g$. IL-6, IL-8, IL-1β, and TNF-α were measured according to the operating instructions of the ELISA kit (Enzyme Immunoassay Industry Company, Jiangsu, China). A standard curve was plotted and the concentration of each treatment group was calculated.

## Flow cytometric analysis

The levels of reactive oxygen species (ROS) were measured using the DCFH-DA indicator (Thermo Fisher Scientific, Rockford, MA, USA). The cell culture supernatant was discarded and cells were suspended in 500 μL binding buffer. Then, 5 μL DCFH-DA indicator was added and cells were incubated at 37˚C for 15 min in the dark. The activity of ROS in cells was measured using flow cytometry.

## MDA analysis

The activity of MDA in cell culture supernatant was measured following the steps of the malondialdehyde assay kit instructions (Nanjing Jiancheng Biotechnology Institute, Nanjing, China). Three replicates were set for each group, and a standard curve was plotted to calculate the activity of MDA.

**Table 2. The primers for qPCR.**

| Gene | Primer sequences (5'-3') | Product length |
|---|---|---|
| IL-1β | F: ATGAAGAGCTGCATCCAACACCTG | 110bp |
| | R: ACCGACACCACCTGCCTGAAG | |
| TNF-α | F: CTGGCGGAGGAGGTGCTCTC | 85bp |
| | R: GGAGGAAGGAGAAGAGGCTGAGG | |
| IL-6 | F: CACTGACCTGCTGGAGAAGATGC | 115bp |
| | R: CCGAATAGCTCTCAGGCTGAACTG | |
| IL-8 | F: AGTGGGCCACACTGTGAAAA | 113bp |
| | R: CCCACAGTACATACATGAGGCA | |
| β-actin | F: CGTCCGTGACATCAAGGAGAAGC | 143bp |
| | R: GGAACCGCTCATTGCCGATGG | |

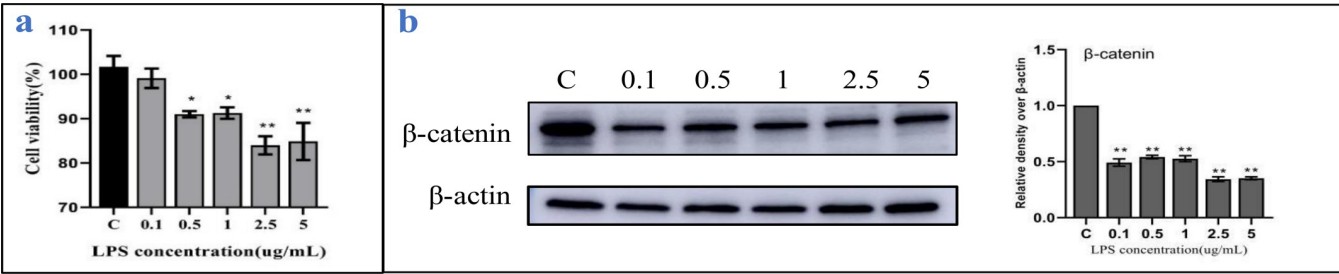

**Fig 1. LPS inhibits the proliferation and expression of β-catenin in BMECs.** BMECs were treated with 0–5 µg/mL LPS for 24 h. (a) The cellular activity of LPS-stimulated BMECs at 24 h was assayed using the CCK-8 method. (b) Effect on the expression of β-catenin in LPS-stimulated BMECs. Western blotting was used to determine the relative levels of β-catenin. β-actin was used as a control. Data are expressed as mean ± SD of 3 independent experiments. $^*0.01 < P < 0.05$, $^{**}P < 0.01$.

### Statistical analysis

Data are expressed as the mean ± standard deviation (SD). Each experiment was repeated 3 times independently. Statistical testing was carried out using $t$-tests for 2 samples or ANOVA for multiple samples. $P > 0.05$, indicated that the difference was not statistically significant; $0.01 < P < 0.05$ indicated that the difference was significant; and $P < 0.01$ indicated that the difference was extremely significant.

## Results and discussion

### Cytotoxicity of LPS on BMECs and inhibition of expression of β-catenin

We used the CCK-8 method to examine the effects of LPS on the biological activity of BMECs. We found that LPS had a significant ($P < 0.05$) inhibitory effect on BMECs in a concentration-depended manner (Fig 1A). For instance, the viability of BMECs stimulated with 0.5 µg/mL LPS was 90%. Western blot analysis revealed that LPS significantly ($P < 0.05$) inhibited the expression of β-catenin in BMECs (Fig 1B). Hence, we stimulated BMECs using 0.5 µg/mL LPS for 24 h to establish a model of cellular inflammation.

### Selection of experimental concentrations in AS-IV- and APS-stimulated BMECs

To investigate the optimal intervention concentration of APS and AS-IV on BMECs, we performed CCK-8 assays to explore cell viability. It should me mentioned that the safe concentrations of these 2 drugs in BMECs differ. We observed that AS-IV at concentrations of 50, 75, and 100 µg/mL (Fig 2A), whereas APS at concentrations of 0.1, 0.25, 0.5, and 1 mg/mL (Fig 2B) did not inhibit the proliferation of BMECs. We also found that the survival rates of BMECs were 93% and 91% when cells were stimulated with 150 and 200 µg/mL AS-IV. Whereas, cell survival was 95% when BMECs were stimulated with 2 mg/mL APS. Accordingly, we selected the 50, 75, and 100 µg/mL AS-IV and 1 mg/mL APS as the standards for subsequent experiments.

### Screening siRNA

A small interfering RNA (siRNA) can specifically regulate a target gene by interfering with or degrading the mRNA of a target gene [23]. The results of our siRNA screening are shown in Fig 3. We noticed that compared with the negative control-siRNA group, both the 1225-β-catenin-siRNA and 1990-β-catenin-siRNA significantly ($P < 0.01$) inhibited the expression of

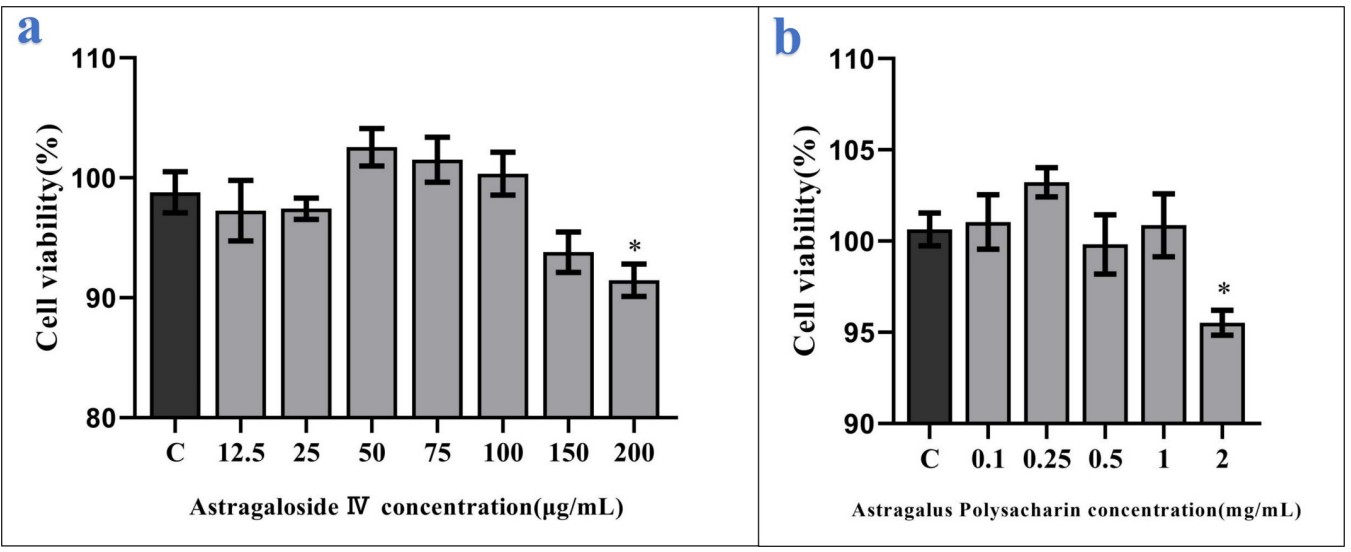

**Fig 2. Influence of AS-IV and APS on the biological activity of BMECs.** The viability of BMECs treated with the indicated concentrations of AS-IV and APS was determined using the CCK-8 assay. Data are expressed as mean ± SD of 6 independent experiments. $^*0.01 < P < 0.05$, $^{**}P < 0.01$.

β-catenin in cells. Therefore, the 1225-β-catenin-siRNA was used as a small interfering RNA in subsequent experiments.

### Influence of AS-IV and APS on Wnt/β-catenin signaling in BMECs

**AS-IV and APS stimulated the activation of the Wnt/β-catenin signaling in BMECs.**
To determine the effects of APS and AS-IV on Wnt signaling in BMECs, we initially examined the changes in the levels of protein in Wnt/β-catenin signaling under normal and inflammatory states in APS- and AS-IV-stimulated BMECs. Subsequently, we examined whether APS or AS-IV contributed to any changes in the levels of protein in Wnt/β-catenin signaling following silencing of β-catenin. We found that stimulation of BMECs with LPS suppressed the expression of β-catenin and cyclin D1 in cells, whereas upregulated the expression of phosphorylated GSK3β (Fig 4B). In addition, we noticed that AS-IV not only antagonized the inhibitory effect of LPS on the Wnt/β-catenin signaling pathway in BMECs (Fig 4B), but also activated the expression of β-catenin, transcription factor 4 (TCF-4), and cyclin D1, inhibited

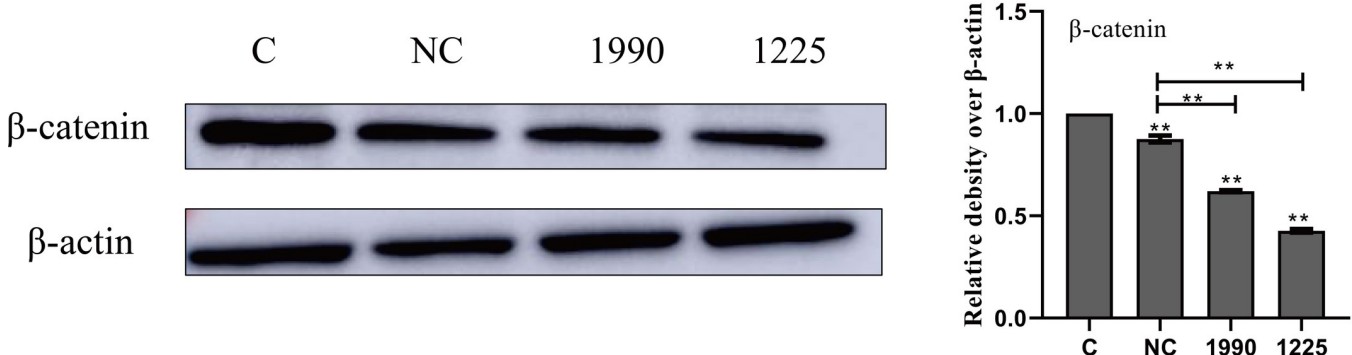

**Fig 3. Effect of siRNA on the expression of β-catenin in BMECs.** C is negative control group; NC is NC-siRNA group; 1225 is 1225-β-catenin -siRNA group; 1990 is 1990-β-catenin-siRNA group. Western blotting was used to determine the relative levels of β-catenin. β-actin was used as a control. Data are expressed as mean ± SD of 3 independent experiments; $^*$ represents $0.01 < P < 0.05$ and $^{**}$ represents $P < 0.01$.

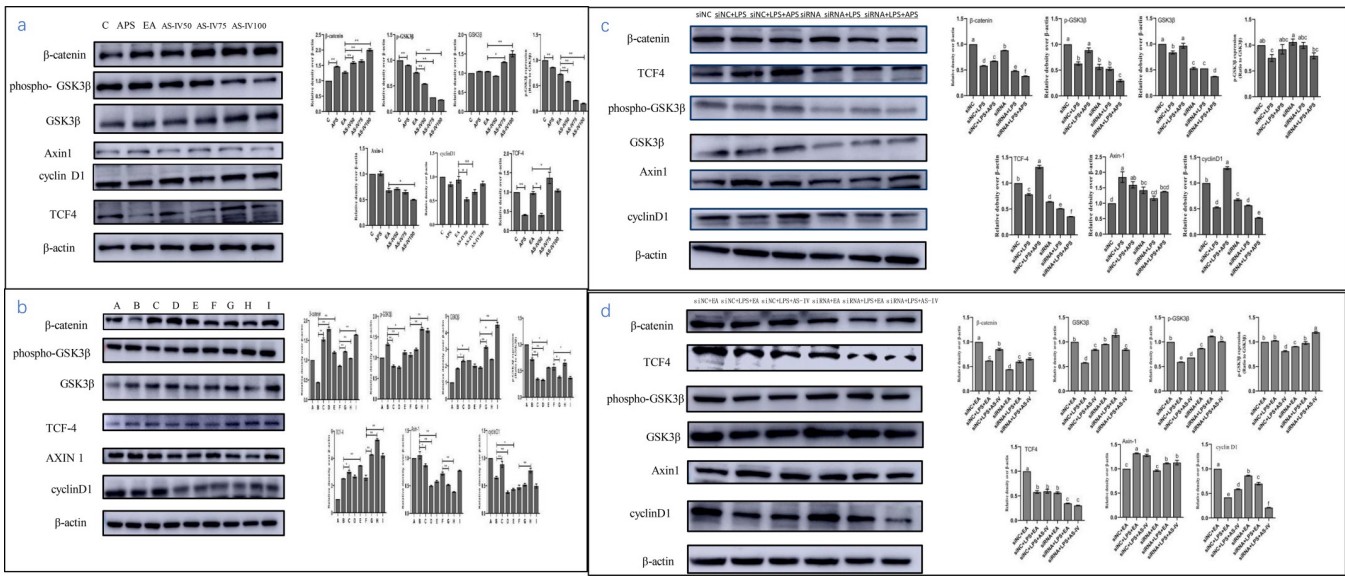

**Fig 4. APS and AS-IV regulation of the Wnt/β-catenin signaling pathway in BMECs.** (a) APS and AS-IV activate the Wnt/β-catenin signaling in BMECs. C = Control cells without any processing; APS = cells treated with 1 mg/mL APS; EA = negative control for AS-IV (ethanol concentration was the same as that in AS-IV 100 μg/mL, ethanol concentration 1%); AS-IV50 = cells treated with 50 μg/mL AS-IV; AS-IV75 = cells treated with 75 μg/mL AS-IV; AS-IV100 = cells treated with 100 μg/mL AS-IV. (b) APS and AS-IV activate Wnt/β-catenin signaling in inflammatory state BMECs. A = Control cells without any processing; B = cells treated with LPS (0.5 μg/mL); C = LPS (0.5 μg/mL) + APS (1 mg/mL); D = LPS (0.5 μg/mL) + lincomycin (50 μg /mL); E = LPS (0.5 μg/mL) + penicillin (86U/mL≈50 μg/mL); F = LPS (0.5 μg/mL) + ethanol concentration 1% (vol/vol); G = LPS (0.5 μg/mL) + AS-IV(50 μg/mL); H = LPS (0.5 μg/mL) + AS-IV(75 μg/mL); I = LPS (0.5 μg/mL) + AS-IV(100 μg/mL). (c, d) Silencing of β-catenin abolishes the ability of APS and AS-IV to regulate the Wnt/β-catenin signaling pathway. siNC = cells treated with NC-siRNA alone; siNC + LPS = NC-siRNA+ LPS (0.5 μg/mL); siNC + LPS + APS = NC-siRNA + LPS (0.5 μg/mL) + APS (1 mg/mL); siRNA = cells treated with 1225-siRNA alone; siRNA + LPS = 1225-siRNA + LPS (0.5 μg/mL); siRNA + LPS + APS = 1225-siRNA+LPS (0.5 μg/mL) + APS (1 mg/mL). siNC + EA = NC-siRNA+ ethyl alcohol (1% vol/vol); siNC + LPS + EA = NC-siRNA + LPS (0.5 μg/mL) + ethyl alcohol (1% vol/vol); siNC + LPS + AS-IV = NC-siRNA+LPS (0.5 μg/mL) +100 μg/mL AS-IV (100 μg/mL); siRNA + EA = 1225-siRNA+ ethyl alcohol (1% vol/vol); siRNA + LPS + EA = 1225-siRNA+LPS (0.5 μg/mL) + ethyl alcohol (1% vol/vol); siRNA + LPS + AS-IV = 1225-siRNA + LPS (0.5 μg/mL) +100 μg/mL AS-IV (100 μg/mL). Protein blots were used to determine relative protein expression levels. β-actin was used as a control. Data are expressed as mean ± SD of 3 independent experiments. * represents $0.01 < P < 0.05$, ** represents $P < 0.01$ and letter differences in the graph indicate significant differences between groups, $P < 0.05$.

the expression of phosphorylated GSK3β and axin-1, and activated the Wnt/β-catenin signaling in untreated (control) BMECs (Fig 4A). We further found that APS also antagonized the LPS-induced inhibition of the Wnt/β-catenin signaling (Fig 4B). Moreover, APS activated the expression of β-catenin in untreated (control) cells but inhibited that of phosphorylated GSK3β, and TCF-4 (Fig 4A). These findings demonstrated that both APS and AS-IV activated the Wnt/β-catenin signaling pathway in BMECs under normal and inflammatory conditions.

**Silencing of β-catenin ameliorated the regulatory role of AS-IV and APS on Wnt/β-catenin signaling.** By comparing the siNC with the siRNA group (Fig 4C) and the siNC+EA with the siRNA+EA group (Fig 4D), we found that 1225-siRNA inhibited the intracellular expression of β-catenin, TCF-4, and cyclin D1 and activated the expression of axin-1 in BMECs, indicating that the Wnt/β-catenin signaling pathway was effectively inhibited by 1225-siRNA. In addition, by comparing the siNC with the siNC+LPS group (Fig 4C) and the siNC+EA with the siNC+LPS+EA group (Fig 4D), we showed that the addition of NC-siRNA to BMECs inhibited the LPS-induced activation of the Wnt/β-catenin signaling in cells, and found that APS and AS-IV antagonized the inhibitory effect of LPS on Wnt/β-catenin signaling. Moreover, we observed that when β-catenin was silenced in BMECs using 1225-siRNA, the ability of APS and AS-IV to activate β-catenin, TCF-4, and cyclin D1 in LPS-stimulated cells was abolished. This finding demonstrated that silencing of β-catenin ameliorated the regulatory effect of APS and AS-IV on the Wnt signaling pathway in cells.

## APS and AS-IV intervention on the levels of IL-6, IL-8, IL-Iβ, and TNF-α inflammatory factors in LPS-stimulated BMECs

First, we tested the effects of APS and AS-IV interventions on the levels of inflammatory factors in LPS-stimulated BMECs at both the transcriptional and translational level using RT-qPCR (Fig 5A) and ELISA (Fig 5B) assays. Our results of the RT-qPCR assay were consistent with the changes observed in the ELISA assays, which showed that LPS-stimulated cells significantly ($P<0.05$) activated the expression of the IL-6, IL-8, IL-1β, and TNF-α inflammatory factors in cells. We noticed that APS and AS-IV significantly ($P<0.05$) downregulated the LPS-induced relative expression of IL-6, IL-8, and TNF-α in BMECs, but had no significant ($P>0.05$) effect on the expression of IL-1β (Fig 5A and 5B). Subsequently, we examined the effect of silencing β-catenin on the expression of IL-6, IL-8, IL-1β, and TNF-α in APS- and AS-IV-treated cells (Fig 5C and 5D). We also found that the addition of NC-siRNA to LPS-stimulated BMECs activated the secretion of IL-6, IL-8, and IL-1β, but had no significant effect on the secretion of TNF-α. Whereas the addition of NC-siRNA, we observed that APS antagonized the secretion of IL-6, IL-8, IL-1β, and TNF-α in LPS-stimulated cells and AS-IV antagonized the secretion of IL-1β and TNF-α in LPS-stimulated cells. Finally, we detected that when β-catenin was silenced (addition of 1225-β-catenin-siRNA), the secretion of IL-6, IL-8, IL-1β, and TNF-α in LPS-stimulated BMECs was ameliorated, and consequently APS and AS-IV did not inhibit the secretion of IL-6, IL-8, IL-1β, and TNF-α in LPS-stimulated BMECs.

## APS and AS-IV interventions on the levels of ROS and MDA oxidative factors in LPS-stimulated BMECs

During cellular injury, an imbalance between oxidative and antioxidant systems in the cell, accompanied by excessive levels of ROS, alters the structure and function of cellular proteins and lipids, leading to cellular dysfunction, which is referred to as oxidative stress [24]. We found that LPS significantly ($P<0.01$) increased the accumulation of ROS in cells, whereas APS significantly ($P<0.05$) inhibited this accumulation in inflammatory cells (Fig 6A). A comparison of the siNC with the siRNA group (Fig 6B) and that of the siNC+EA with the siRNA+EA group (Fig 6C) revealed that ROS was significantly ($P<0.05$) accumulated in cells after silencing β-catenin. Following silencing of β-catenin, we found that LPS inhibited the accumulation of ROS in cells, while AS-IV had no inhibitory effect on ROS in inflammatory cells; whereas APS inhibited the secretion of ROS in inflammatory cells.

Malondialdehyde (MDA) is the end product of lipid oxygen radicals in cells. It is cytotoxic and can directly reflect the degree of lipid peroxidation and indirectly the content of oxygen radicals in cells and the degree of cell damage [25]. Our results of the MDA assay are shown in Fig 7. We observed that LPS stimulation of BMECs significantly ($P<0.01$) upregulated the levels of MDA in cells, whereas APS and AS-IV antagonized the LPS-induced secretion of MDA (Fig 7A). When NC-siRNA was added, we detected that LPS upregulated the levels of MDA in BMECs, causing cellular damage, whereas APS and AS-IV exerted an inhibitory effect on the levels of MDA in inflammatory cells. We further found that silencing of β-catenin resulted in the loss of the ability of LPS to upregulate MDA, as well as that of APS and AS-IV to regulate the levels of MDA in inflammatory cells.

## Conclusions

Mastitis in dairy cows is caused by the invasion of pathogenic bacteria or toxins that result in persistent inflammation in the mammary tissue. BMECs are immunocompetent, lactogenic, and highly differentiated cells that constitute the first line of defense of the breast tissue against

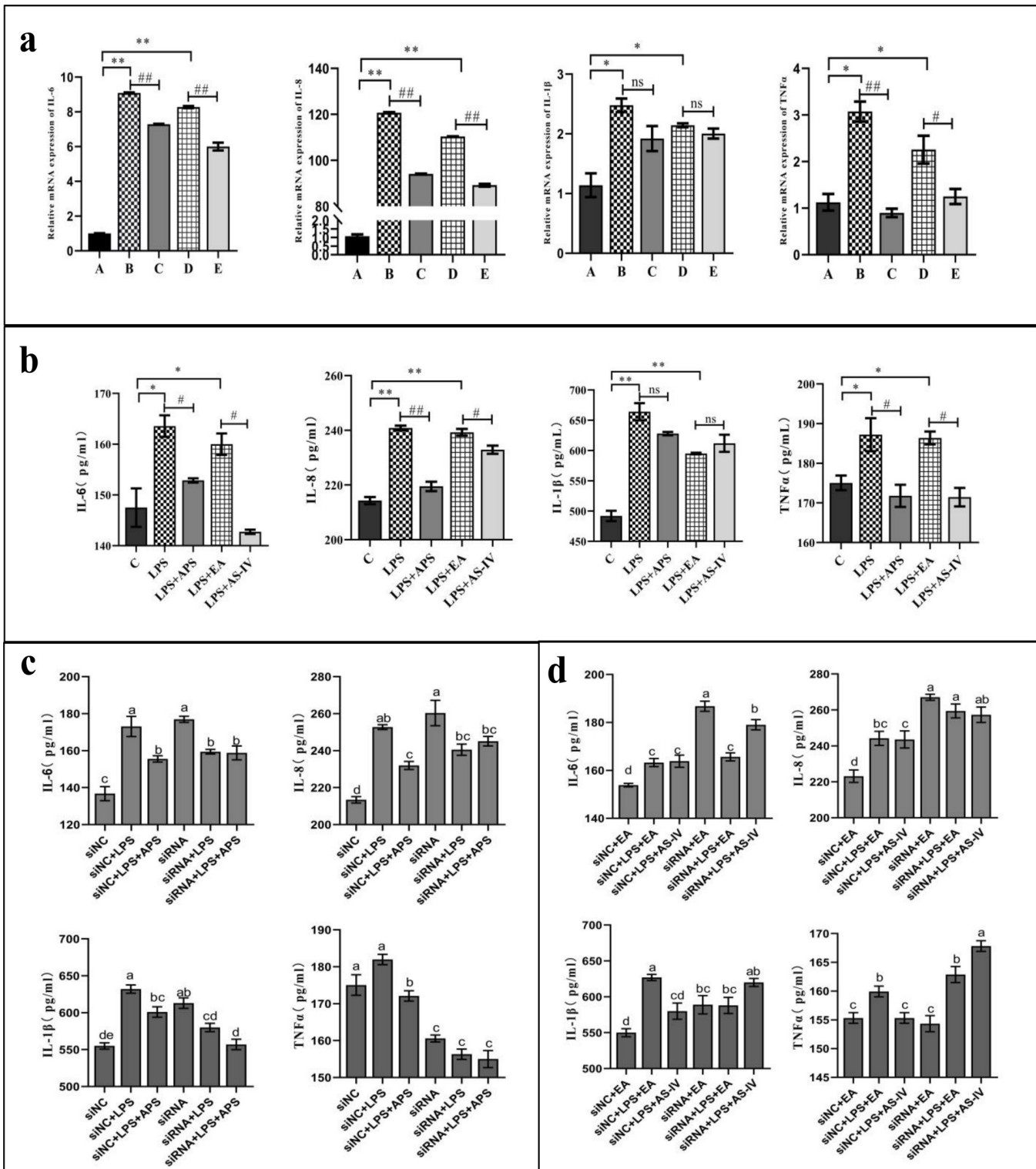

**Fig 5. APS and AS-IV inhibit the expression of inflammatory factors in BMECs under an inflammatory state.** (a, b) APS and AS-IV suppress the expression of inflammatory factor genes and proteins in LPS-stimulated BMECs. C = Control cells without any processing; LPS = cells treated with LPS (0.5 μg/mL) alone; LPS + APS = LPS (0.5 μg/mL) + APS (1 mg/mL); LPS + EA = LPS (0.5 μg/mL) + ethanol concentration 1% (vol/vol); LPS + AS-IV = LPS (0.5 μg/mL) + AS-IV (100 μg/mL). (c, d) APS and AS-IV fail to inhibit inflammatory factors in BMECs under an inflammatory state after silencing β-catenin. Each experimental group expresses the same meaning as in Fig 4C and 4D. Data are expressed as the mean ± SD of 3 independent experiments. *, # represents 0.01 < $P$ < 0.05, **, ## represents $P$ < 0.01, ns represents $P$ > 0.05, and letter differences in the graphs indicate significant $P$ < 0.05 differences between groups.

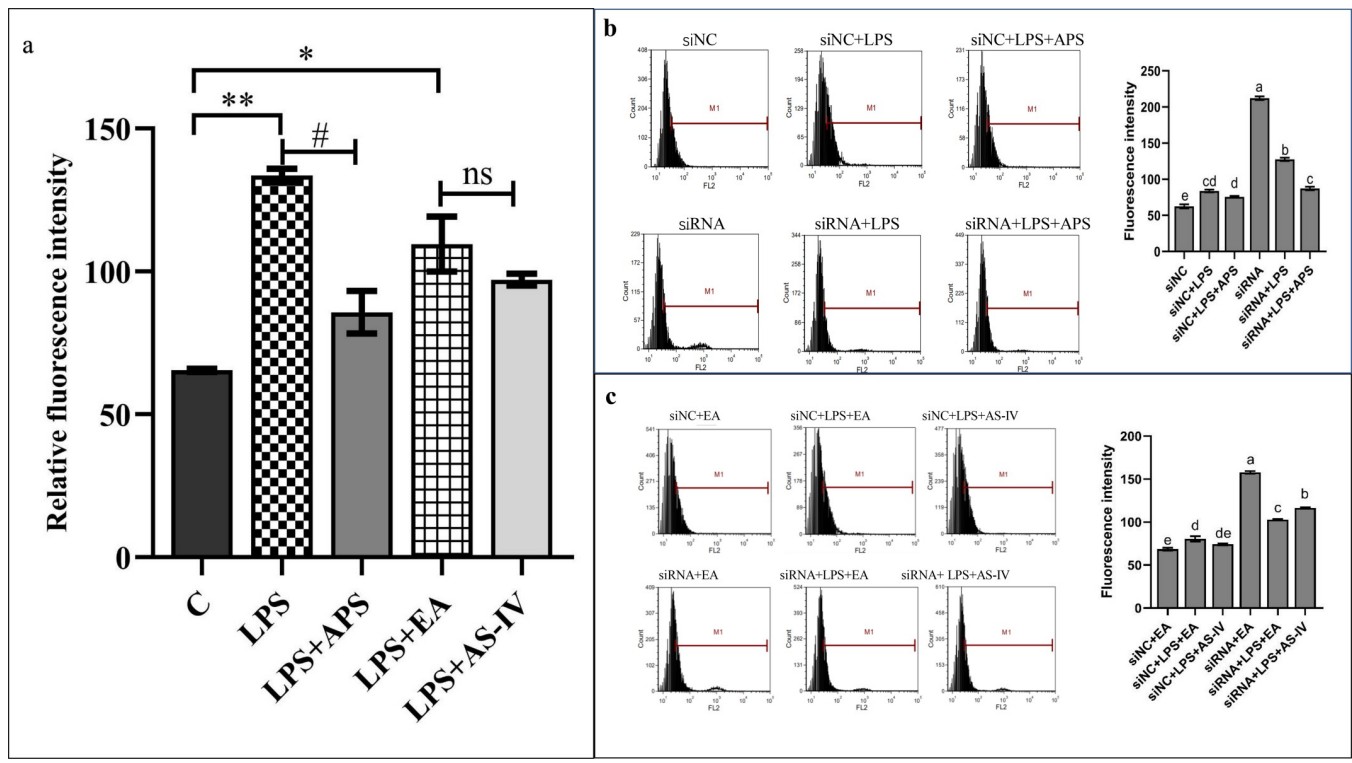

**Fig 6. APS and AS-IV regulate the accumulation of ROS in BMECs under an inflammatory state.** (a) APS and AS-IV inhibit the LPS-stimulated accumulation of ROS in LPS-stimulated BMECs. (b, c) Effect of APS and AS-IV on the accumulation of ROS in LPS-stimulated BMECs after silencing β-catenin. Each experimental group expresses the same meaning as in Fig 5. Data are expressed as the mean ± SD of 3 independent experiments. *, # represents $0.01 < P < 0.05$, **, ## represents $P < 0.01$, ns represents $P > 0.05$, and letter differences in the graphs indicate significant $P < 0.05$ differences between groups.

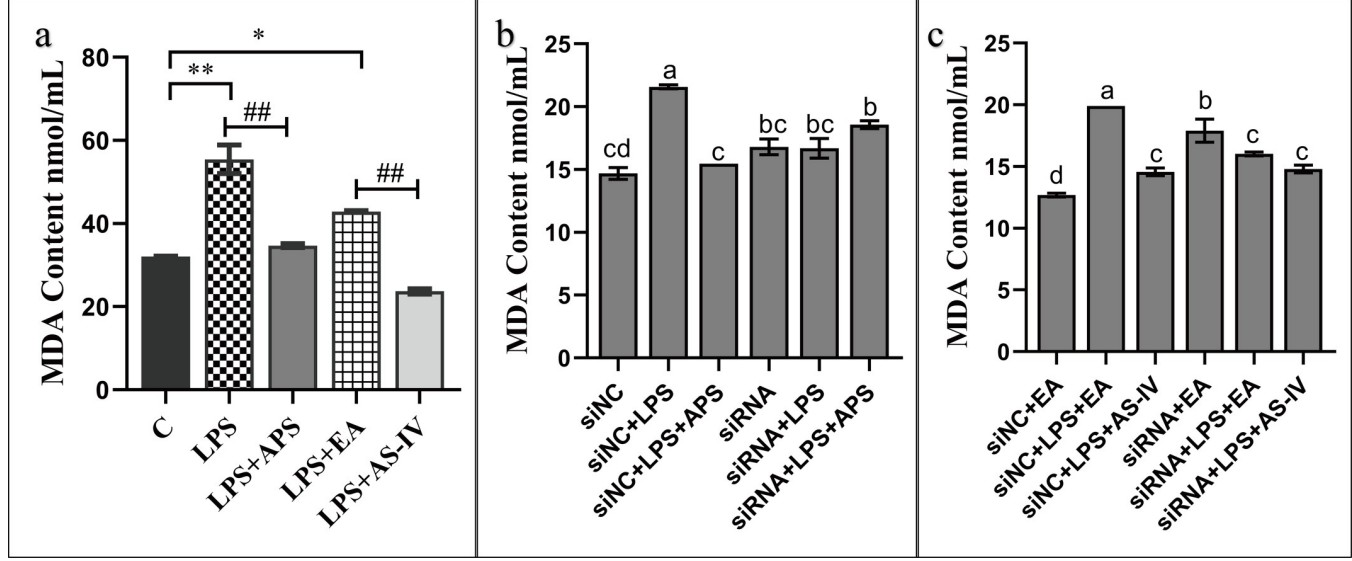

**Fig 7. APS and AS-IV regulate the levels of MDA in BMECs under an inflammatory state.** (a) APS and AS-IV inhibit the LPS-stimulated upregulation of MDA in LPS-stimulated BMECs. (b, c) Effect of APS and AS-IV on the levels of MDA in LPS-stimulated BMECs after silencing β-catenin. Each experimental group expresses the same meaning as in Fig 5. Data are expressed as the mean ± SD of 3 independent experiments. *, # represents $0.01 < P < 0.05$, **, ## represents $P < 0.01$, ns represents $P > 0.05$, and letter differences in the graphs indicate significant $P < 0.05$ differences between groups.

invasion by harmful external agents [5]. In addition, BMECs are the main cell population in the mammary tissue of dairy cows, and their cellular status directly influences lactation capacity and mammary gland health [26]. LPS, a major component of the cell wall of *E. coli*, is one of the main factors known to exacerbate mastitis [27]. Stimulation of BMECs or murine mammary tissues with LPS or *E. coli* caused similar infection conditions and clinical features, with LPS showing better reproducibility in stimulating the generation of mastitis [28, 29]. Therefore, LPS has become an ideal agent for generating mastitis models.

LPS is the main substance that induces inflammation and affects the cellular activity of BMECs in a time- and dose-dependent manner [5]. Based on previous studies in our laboratory, we chose the smallest concentration causing inflammation and avoiding significant cell damage [30]. We thus established a model of inflammation by stimulating BMECs with 0.5 μg/mL LPS for 24 h. Accordingly, in our LPS-stimulated BMEC inflammation models, we found the increased expression of IL-6, IL-8, IL-1β, and TNF-α inflammatory factors, as well as that of ROS oxidation factors in inflammatory cells, which was consistent with previous studies [4, 30, 31].

The Wnt signaling mediates cellular self-renewal and repair in a variety of epithelial cells and is essential for the development of tissues and organs [32–34]. An increasing number of studies have shown that inflammatory responses, oxidative stress, and disruption of the Wnt/β-catenin signaling pathway are associated with a variety of pathological processes. For example, the activity of Wnt signaling is increased in diseases such as cancer [35], rheumatoid arthritis [36], lupus erythematosus [37], and systemic sclerosis [14], whereas, it is inhibited in a variety of other diseases, such as diabetic foot ulcers [38] and cleft palate [39]. Several studies have reported the controversial role of β-catenin as an inflammatory modulator [9, 10]. Here, we showed that the activity of the Wnt/β-catenin signaling pathway was inhibited in LPS-induced BMECs (Fig 4B).

In this study, we experimentally demonstrated that β-catenin in Wnt signaling negatively regulated the expression of inflammation and oxidative stress factors in LPS-treated BMECs, similar to the findings by Jiang et al. [10, 40, 41]. In experiments of bovine endometrial epithelial cells stimulated with LPS, LPS was reported to inhibit the intracellular expression of β-catenin and promote inflammatory cell damage [40]. Moreover, β-catenin was found to negatively regulate the inflammatory response induced by *Salmonella typhimurium*, and stimulation of mouse models and colonic epithelial cells by this strain aided the degradation of β-catenin and increased the expression of IL-6 and TNF-α [10]. LPS produced by *Porphyromonas gingivalis* inhibited the activity of the Wnt/β-catenin signaling pathway in human periodontal ligament cells (hPDLCs), activated the cellular inflammatory response, and inhibited the differentiation of hPDLCs [41].

Antibiotics are often used to kill pathogenic microorganisms in the mammary tissues of cows, following the occurrence of mastitis. However, most pathogens are highly resistant to treatment due to a lack of awareness regarding the misuse of antibiotics [42]. In addition, antibiotics promote the release of large amounts of bacterial compounds that enhance inflammation, further limiting the effect of antibiotics in reducing inflammation [11]. Hence, it is important to identify natural products and understand their targets of action in eliminating inflammation as an alternative to antibiotics in the treatment of mastitis. Both, APS and AS-IV, which are abundant in nature and possess various biological activities, are active ingredients extracted from RA [19]. In our study, we used the BMEC inflammation model to explore the mechanisms of action of APS and AS-IV. Our results revealed that APS and AS-IV played a protective role in reducing the secretion of the IL-6, IL-8, and TNF-α inflammatory factors, as well as that of the MDA oxidative factor in LPS-induced injury in BMECs (Figs 5A, 5B and 7A), with APS also inhibiting the accumulation of ROS in inflammatory cells (Fig 6A).

APS and AS-IV have the ability to regulate cell proliferation and inflammation by modulating the Wnt/β-catenin signaling pathway [16, 18]. We investigated the effects of APS and AS-IV interventions on LPS-induced changes in the Wnt/β-catenin signaling pathway in BMECs. When BMECs under a normal or inflammatory state were stimulated with safe concentrations of APS and AS-IV, the Wnt/β-catenin signaling pathway was activated (Fig 4A and 4B). When 1225-siRNA was used to silence the expression of β-catenin, the ability of APS and AS-IV to inhibit the IL-6, IL-8, TNF-α inflammatory factors, and the MDA oxidative factor in LPS-stimulated BMECs was abolished (Figs 5C, 5D, 7B and 7C). One possible explanation of this antiinflammatory mechanism might be the fact that APS and AS-IV antagonized for the LPS-induced inhibition of Wnt/β-catenin signaling. The regulation of β-catenin by APS and AS-IV remains also controversial. LiCl was found to upregulate the expression of β-catenin in mouse keratin-forming cells and inhibit their proliferation and migration, whereas AS-IV antagonized this effect and enhanced cell proliferation [16]. Furthermore, APS activated the expression of Wnt2, β-catenin, and LRP5 genes and inhibited oxidative stress in deovulated rats to reduce osteoporosis [43]. Finally, APS inhibited the Wnt/β-catenin signaling pathway in breast cancer cells to reduce cell proliferation and epithelial mesenchymal transition (EMT)-mediated cell migration and invasion [18]. Thus, the Wnt signaling can be considered as one of the targets of APS and AS-IV for inhibiting inflammation.

In summary, our results demonstrated that the Wnt/β-catenin signaling pathway is a target of APS and AS-IV for protection against LPS-induced inflammatory injury in BMECs.

## Supporting information

**S1 Raw images.**
(PDF)

## Acknowledgments

The authors would like to thank Editage (www.editage.cn) for their assistance with English language editing.

## Author Contributions

**Data curation:** Jiaqi Fan.

**Formal analysis:** Jiaqi Fan.

**Funding acquisition:** Fang Jia, Xuezhang Zhou.

**Investigation:** Fang Jia.

**Methodology:** Fang Jia.

**Supervision:** Xuezhang Zhou.

**Writing – original draft:** Jiaqi Fan.

**Writing – review & editing:** Yang Liu, Xuezhang Zhou.

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
