## [Decision Letter · Decision Letter 0]

27 Apr 2022

PONE-D-22-02294Astragalus Polysaccharides and Astragaloside IV alleviate Inflammation in Bovine Mammary Epithelial Cells by regulating Wnt/β-catenin Signaling PathwayPLOS ONE

Dear Dr. Zhou,

Thank you for submitting your manuscript to PLOS ONE. After careful consideration, we feel that it has merit but does not fully meet PLOS ONE’s publication criteria as it currently stands. Therefore, we invite you to submit a revised version of the manuscript that addresses the points raised during the review process.

PLEASE ADDRESS THE COMMENTS AND CONCERNS RAISED BY THE REVIEWER.

We look forward to receiving your revised manuscript.

Kind regards,

Juan J Loor

Academic Editor

PLOS ONE

Journal Requirements:

Reviewers' comments:

Reviewer's Responses to Questions

**Comments to the Author**

1. Is the manuscript technically sound, and do the data support the conclusions?

Reviewer #1: Yes

2. Has the statistical analysis been performed appropriately and rigorously? 

Reviewer #1: Yes

3. Have the authors made all data underlying the findings in their manuscript fully available?

Reviewer #1: Yes

4. Is the manuscript presented in an intelligible fashion and written in standard English?

Reviewer #1: Yes

5. Review Comments to the Author

Reviewer #1: "Astragalus Polysaccharides and Astragaloside IV alleviate Inflammation in Bovine Mammary Epithelial Cells by regulating Wnt/β-catenin Signaling Pathway" aimed to understand the mechanisms by which Astragalus polysaccharides (APS) and astragaloside IV (AS-IV) interact with Wnt signaling to achieve therapeutic effects in bovine mammary epithelial cells (BMECs). The authors concluded that APS and AS-IV antagonized the inhibitory effect of LPS. They mediate protective effects in the inflammatory BMECs model by activating the Wnt signaling pathway.

The last part of the abstract (5 or 6 last sentences) is hard to follow, and it lacks a link between sentences. It needs to be improved to offer a smooth or cohesive connection between the topics addressed.

The Results and Discussion section only presents the results, and there is no discussion at all. Moreover, the Conclusion section is more like a Discussion section. This manuscript needs significant work on the results and discussion sections to tell the story and the benefits of the findings (if applicable). Furthermore, it would be interesting if the authors address how this in vitro finding could be applicable in vivo.

Some line-by-line comments:

Line 9: Add a comma after repair. Remember to use oxford comma.

Line25: add the before Wtn.

Line 37: remove the comma between year and by.

Line 42: remove the comma between stress and establish.

Line 44 – 47: hard to read sentence, it is too long, consider breaking it into two sentences.

Line 46: it should read have instead of has.

Line 67 – 68: rephrase “The aim of this study was to investigate” for “This study aimed to …”

Line 70 – 71: take out this sentence “Here, we demonstrated that APS and AS-IV exerted cytoprotective effects on BMECs through the negative regulation of inflammation and Wnt/β-catenin signaling.” Results should not be in the Introduction section.

Line 74: do not start a sentence with the abbreviation, fully spell the word.

Line 75 do not start a sentence with the abbreviation, fully spell the word.

Line 82: do not start a sentence with the abbreviation, fully spell the word.

Line 84: do not start a sentence with the abbreviation, fully spell the word. I will stop these comments here, check the whole manuscript and make the pertinent changes.

Line 88: at what temperature were cells incubated?

Line 91: at what temperature were cells incubated?

Line 93: at what temperature were cells incubated?

Line 94: at what temperature were cells incubated?

Line 103 – 107: hard to read sentence, it is too long, consider breaking it into two sentences.

Line 151 -153: hard to read sentence, rephrase it.

Line 171:it should read be instead of me.

6. PLOS authors have the option to publish the peer review history of their article (what does this mean?). If published, this will include your full peer review and any attached files.

Reviewer #1: No

---

## [Author Response · Author response to Decision Letter 0]

12 Jun 2022

Dear Editor:

Thank you very much for your email with which you sent us the reviewer’s report on our paper with the reference number PONE-D-22-02294. We also wish to take this opportunity to thank the reviewer for his constructive comments and valuable recommendations. We have carefully revised the manuscript according to reviewer’s suggestion.

Our responses to several comments are listed below:

Comment1: "Astragalus Polysaccharides and Astragaloside IV alleviate Inflammation in Bovine Mammary Epithelial Cells by regulating Wnt/β-catenin Signaling Pathway" aimed to understand the mechanisms by which Astragalus polysaccharides (APS) and astragaloside IV (AS-IV) interact with Wnt signaling to achieve therapeutic effects in bovine mammary epithelial cells (BMECs). The authors concluded that APS and AS-IV antagonized the inhibitory effect of LPS. They mediate protective effects in the inflammatory BMECs model by activating the Wnt signaling pathway.

The last part of the abstract (5 or 6 last sentences) is hard to follow, and it lacks a link between sentences. It needs to be improved to offer a smooth or cohesive connection between the topics addressed.

Reply: The 6 last sentence has been changed to increase its connection with the previous sentence.

Comment2: The Results and Discussion section only presents the results, and there is no discussion at all. Moreover, the Conclusion section is more like a Discussion section. This manuscript needs significant work on the results and discussion sections to tell the story and the benefits of the findings (if applicable). Furthermore, it would be interesting if the authors address how this in vitro finding could be applicable in vivo.

Reply: In the results and discussion we added a little discussion, hoping to give future readers more thinking directions. In addition, it was boldly speculated that AS-IV and APS might be due to the cell activity affecting Wnt signaling, thus reducing the continuation of inflammation. At the end of this paper, we describe the effect of Radix Astragali powder or APS injection by other researchers on the mammary gland of dairy cows. If we want to promote the use of AS-IV and APS in clinical treatment of cow mastitis in the future, we need animal tests to verify the therapeutic effects of AS-IV and APS on mastitis caused by different factors.

Comment3: Line 9: Add a comma after repair. Remember to use oxford comma.

Reply: It has been modified as required.

Comment4: Line25: add the before Wtn.

Reply: I am very sorry that I did not understand this revised opinion. Line 25 has not been modified.

Comment5: Line 37: remove the comma between year and by. Line 42: remove the comma between stress and establish.

Reply: Lines 37 and 42 have been punctuated as required.

Comment6: Line 44 – 47: hard to read sentence, it is too long, consider breaking it into two sentences.

Reply: We have broken down lines 44-47 into two sentences. “Accordingly, both the NF-κB and MAPK signaling pathways are known to be positively associated with cellular inflammation [7-9]. Whereas an increasing number of studies have identified disruptions in the Wnt/β-catenin signaling pathway that accompany inflammatory responses in a variety of diseases [10, 11].”

Comment7: Line 46: it should read have instead of has.

Reply: It has been modified as required.

Comment8: Line 67 – 68: rephrase “The aim of this study was to investigate” for “This study aimed to …”

Reply: It has been modified as required.

Comment9: Line 70 – 71: take out this sentence “Here, we demonstrated that APS and AS-IV exerted cytoprotective effects on BMECs through the negative regulation of inflammation and Wnt/β-catenin signaling.” Results should not be in the Introduction section.

Reply: This sentence was deleted from the preface and included in the discussion.

Comment10: Line 74: do not start a sentence with the abbreviation, fully spell the word.

Line 75 do not start a sentence with the abbreviation, fully spell the word.

Line 82: do not start a sentence with the abbreviation, fully spell the word.

Line 84: do not start a sentence with the abbreviation, fully spell the word. I will stop these comments here, check the whole manuscript and make the pertinent changes.

Reply: We have checked the entire manuscript and changed the abbreviations at the beginning of sentences to full spelling.

Comment11: Line 88: at what temperature were cells incubated?

Line 91: at what temperature were cells incubated?

Line 93: at what temperature were cells incubated?

Line 94: at what temperature were cells incubated?

Reply: Cells were cultured in a 37 °C incubator containing 5% CO2 and 95% AIR. Cell culture temperature was added in lines 88, 91, 93 and 94.

Comment12: Line 103 – 107: hard to read sentence, it is too long, consider breaking it into two sentences.

Reply: We have broken down lines 103-107 into two sentences. “PVDF membranes were incubated with β-catenin, GSK3β, phospho-GSK3β (Bioss, Beijing, China), axin-1, TCF-4, cyclinD1 (Biosciences, Shanghai, China), and β-actin (Abcam, Cambridge, UK) antibodies for 12 h at 4 °C and washed 3 times with TBST. PVDF membranes were incubated with horseradish peroxidase-labeled secondary antibodies (Abcam) for 2 h at 25 °C and a final thrice wash with TBST.”

Comment13: Line 151 -153: hard to read sentence, rephrase it.

Reply: We have broken down lines 151-153 into three sentences. “P > 0.05 was not statistically significant. 0.01 <P < 0.05 was a significant difference. P < 0.01 indicates extremely significant difference.”

Comment14: Line 171:it should read be instead of me.

Reply: “It should me mentioned that the safe concentrations of these 2 drugs in BMECs differ.” This sentence is amended “The safe concentrations of the two drugs in BMECs are different.”

---

## [Decision Letter · Decision Letter 1]

4 Jul 2022

Astragalus Polysaccharides and Astragaloside IV alleviate Inflammation in Bovine Mammary Epithelial Cells by regulating Wnt/β-catenin Signaling Pathway

PONE-D-22-02294R1

Dear Dr. Zhou,

We’re pleased to inform you that your manuscript has been judged scientifically suitable for publication and will be formally accepted for publication once it meets all outstanding technical requirements.

Kind regards,

Juan J Loor

Academic Editor

PLOS ONE

Additional Editor Comments (optional):

Reviewers' comments:

Reviewer's Responses to Questions

**Comments to the Author**

1. If the authors have adequately addressed your comments raised in a previous round of review and you feel that this manuscript is now acceptable for publication, you may indicate that here to bypass the “Comments to the Author” section, enter your conflict of interest statement in the “Confidential to Editor” section, and submit your "Accept" recommendation.

Reviewer #1: All comments have been addressed

2. Is the manuscript technically sound, and do the data support the conclusions?

Reviewer #1: Yes

3. Has the statistical analysis been performed appropriately and rigorously? 

Reviewer #1: Yes

4. Have the authors made all data underlying the findings in their manuscript fully available?

Reviewer #1: Yes

5. Is the manuscript presented in an intelligible fashion and written in standard English?

Reviewer #1: Yes

6. Review Comments to the Author

Reviewer #1: (No Response)

7. PLOS authors have the option to publish the peer review history of their article (what does this mean?). If published, this will include your full peer review and any attached files.

Reviewer #1: No

---

## [Editor Report · Acceptance letter]

14 Jul 2022

PONE-D-22-02294R1 

Astragalus Polysaccharides and Astragaloside IV alleviate Inflammation in Bovine Mammary Epithelial Cells by regulating Wnt/βß-catenin Signaling Pathway 

Dear Dr. Zhou:

I'm pleased to inform you that your manuscript has been deemed suitable for publication in PLOS ONE. Congratulations! Your manuscript is now with our production department. 

Kind regards, 

on behalf of

Dr. Juan J Loor 

Academic Editor

PLOS ONE